# Uncertainty-Based Autonomous Path Planning for Laser Line Scanners

**Michiel Vlaeyen** [1,2,*], **Han Haitjema** [2] and **Wim Dewulf** [1,2]

1   Member of Flanders Make—Core Lab MaPS, KU Leuven, Celestijnenlaan 300, 3001 Leuven, Belgium
2   Manufacturing Metrology Section, MaPS, KU Leuven, Celestijnenlaan 300, 3001 Leuven, Belgium
*   Correspondence: michiel.vlaeyen@kuleuven.be

**Abstract:** This study proposes an algorithm to autonomously generate the scan path for a laser line scanner mounted on a coordinate measuring machine. The scan path is determined based on task-specific measurement uncertainty in order to prove conformance to specified tolerances. The novelty of the algorithm is the integration of measurement uncertainty. This development is made possible by recent developments for digital twins of optical measurement systems. Furthermore, the algorithm takes all the constraints of this optical measurement system into account. The proposed algorithm is validated on different objects with different surface characteristics. The validation is performed experimentally by a physical measurement system and virtually by an in-house developed digital twin. The validation proves that theoretical coverable areas are measured properly, and the method applied to the equipment used leads to adequate measurement paths that give measurements results with sufficient measurement uncertainty to prove conformance to specifications.

**Keywords:** laser line scanner; path planning; digital twin; virtual CMM

## 1. Introduction

The manufacturing industry evolves towards individualized and lot-size-one products with zero-defect manufacturing [1]. This ongoing trend poses new challenges to the manufacturing industry. In order to competitively address these challenges, the Industry 4.0 (r)evolution is being put forward as a holistic concept and game changer for the sector. Manufacturing metrology has a twofold role to play within this transformation. On the one hand, manufacturing metrology is considered a pacemaker in the Industry 4.0 concept, since metrology provides measurement data both for steering and for validating cyber-physical production systems [2]. As a result, the data collection needs to be fast, accurate and applicable for lot-size-one products [3]. Quality assurance (QA) with contact probes is not applicable for fast data collection [4]. Contact measurements such as roughness measurements [5] or tactile CMM probing [6] are considered slower than non-contact measurements, such as laser line scanning, high-speed cameras [7], 3D laser doppler vibrometry systems [8] and digital image correlation [9]. As a result, the number of applications of non-contact probes has increased within research and industry over the last decade [10]. On the other hand, the manufacturing metrology systems themselves need to evolve towards self-organized and self-optimized measurement systems that can achieve 100% quality assurance of lot-size-one products. Therefore, the scan strategy needs to be determined autonomously, without interference of an operator on the quality of the inspection. Manual QAs often suffer from over- or under-scanning, resulting in a decreased quality or unnecessary time loss [11]. Furthermore, the manually-determined scan paths have no quality indication and are difficult to reproduce [12]. An autonomously generated scan strategy for a QA may exclude user errors. The automated path planning is essential for a transition towards the Industry 4.0.

Firstly, this article describes the optical system used in the research, namely a laser line scanner (LLS) mounted on a coordinate measuring machine (CMM) and its digital twin.

Secondly, an overview of the development of the path planning algorithms over the last years is given. Thirdly, based on the shortcomings of the state-of-the-art algorithms, a new path planning algorithm is proposed. Lastly, the proposed algorithm is validated on an object with multiple tolerant features and on a complex virtual object, in order to show the capabilities of the algorithm.

## 2. Optical Measurement System

The optical measurement system for which the path planning algorithm is created is an LLS mounted on a CMM. The conventional CMM positions the probing system, i.e., the LLS, in the CMM space. This space is depicted in Figure 1 and the boundaries are set by the limits of the three axes of movement of the CMM. The three axes are orthogonally to each other, and they are the coordinate system of the CMM space. While measuring, the optical probing system must move in a straight line. The articulating probe head (aph) orients the LLS by means of two incremental joints, illustrated in Figure 2. Each joint has an increment of 7.5°. Joint A has a minimal angle of 0°, i.e., a downwards orientation, and a maximum angle of 105°, i.e., a slightly upwards orientation. Joint B can be oriented from −180° up to 180°. The articulating probe head works asynchronously with the CMM. As a consequence, the LLS cannot alter its orientation and position simultaneously.

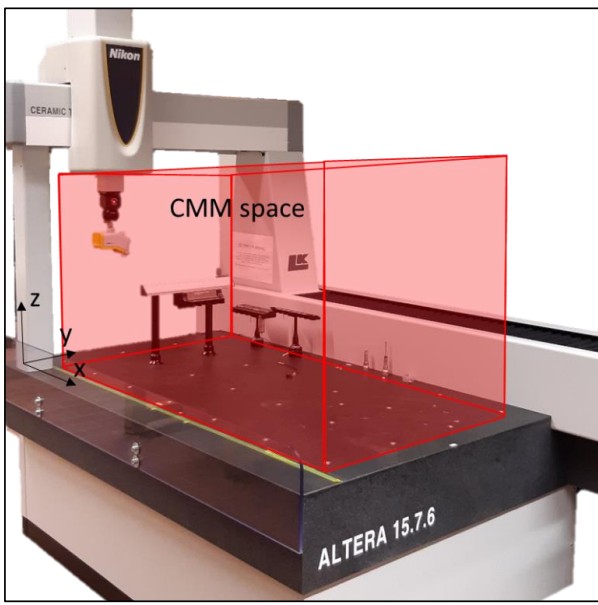

**Figure 1.** The CMM with the CMM space.

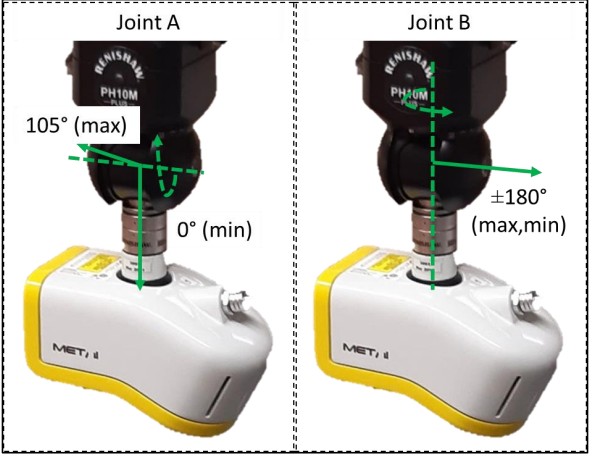

**Figure 2.** The LLS and aph with the limits of both joints.

The probing system is an LLS, depicted in Figure 2. The LLS is a probing system that generates the measured points by making use of triangulation. This probe generates data by projecting a plane of laser light onto the object's surface and the intersection thereof is determined by capturing the reflected laser light. The non-contact character of an optical measurement system, like an LLS, allows for a high measuring speed and a high-density sampling point generation compared with the conventional tactile probing systems [13–15]. On the other hand, the precision can be an order of magnitude worse than a tactile system. As a consequence, the reachability of the surface forms a constraint for the use of the LLS. Additionally, the LLS needs to respect a minimal distance to the object, i.e., the scan depth. This minimal distance is referred to as standoff distance. Furthermore, only the surface within the field of view, defined by the width of view and the depth of view, can be scanned. Lastly, the angle between the laser beam and the normal on the surface must be smaller than the maximum angle of reflectance. This maximal angle of reflectance is not only dependent on the surface characteristics, but also on the laser settings, i.e., laser intensity and exposure time. Since this study assumes uniform surface characteristics and constant laser settings, the maximum angle of reflectance is constant for the algorithm. This value is dependent on the surface characteristics, sensor settings and ambient conditions. Figure 3 illustrates the constraints of the LLS.

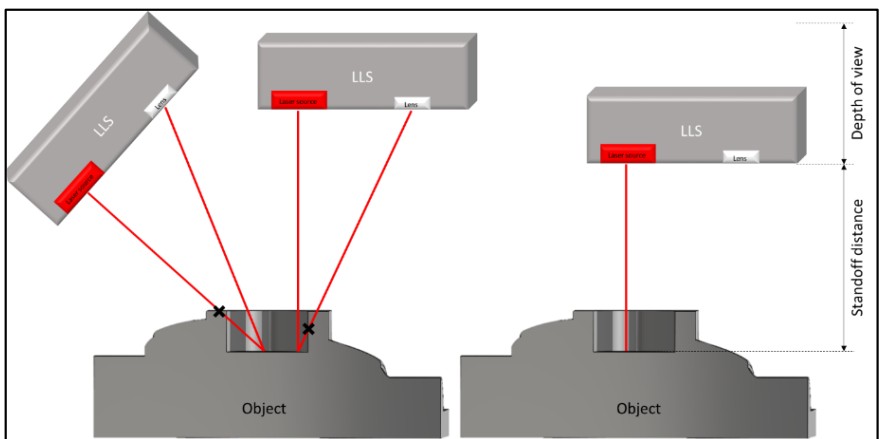

**Figure 3.** The physical constraints of the probing system related to its positioning relative to the measured object. On the left is the reachability constraint and on the right is the scan depth.

The digital twin of the LLS mounted on the CMM can be defined as a model of the measurement system that uses an evolving set of data and adjusts the model in accordance with the required data [16]. In this study, the parameters of the digital twin were chosen in such a manner that it mimics its physical counterpart: the Coord3 MC16 CMM and the Nikon Metrology NV LC60Dx LLS. The development of the used digital twin is discussed in detail in [17]. The digital twin can be used to evaluate the generated scan path. Collisions between the probing system and the measured object can be visualized. All the constraints can be checked by the digital twin. Furthermore, the digital twin can be used to determine the uncertainty. This uncertainty is required to prove conformance to specified tolerances according to International Organization for Standardization (ISO) 14253-1 [18]. The task-specific measurement uncertainty when measuring these tolerant features depends on multiple contributors such as: environmental factors, surface characteristics, the measurements system and the measurement strategy [19,20]. Since the measurement strategy influences the uncertainty budget, an uncertainty estimation performed by the digital twin forms an essential part of an autonomous path planning algorithm.

### 3. Path Planning Algorithms: State-of-the-Art

The state-of-the-art path planning algorithms are based on a priori known information the computer-aided design (CAD) model of the object. The scan path generation algorithms

have resemblance to toolpath planning for computer numerical control (CNC). However, toolpath planning for CNC comes down to driving a point over a curve, while an LLS covers a surface with a line. Tarbox and Gottschlich developed planning algorithms to inspect an object's surface based on an object model [21]. Another path generation algorithm for LLS was developed by F. Xi and C. Shu [11]. This algorithm targets the constraint of the width of the field of view by dividing the CAD model into a number of sections, which can be scanned in one stroke. The drawback of this algorithm is that the depth of the field of view, maximum angle of reflectance and occlusion are not taken into account. The path planning algorithm of K.H. Lee et al. [15,22–25] takes the constraint concerning the maximum angle of reflectance and reachability into account. In order to minimize the computing time, a region-growing algorithm is presented where points on the surface are clustered in regions that can be scanned with the same orientation. The effectiveness and applicability of this path generation algorithm has been proven by L. Ding et al. and F. A. R. Martins [4,26]. Another method to determine feasible viewpoints for the LLS is presented by P. Fernandez et al. [12]. A. Magaña et al. were able to determine the scan path based on feature cluster constrained spaces, taking into account the constraints of optical measurement systems [27]. Other literature discusses the combined approach of an LLS and a tactile probe [28,29]. The combined method allows for complete scanning of the object's reachable surface with the LLS and measurement with higher accuracy using the tactile probe. M. Mahmud et al. presented a method to determine the scan path as a function of the measurement uncertainty in order to confirm the specifications [30]. Recent algorithms focus on the minimization of the inspection time while fully scanning the object [31,32]. This is realized by minimizing the number of orientations and by controlling the overlap between scan paths. Related research discusses path planning for laser scanners positioned by an industrial robot [33,34]. Robot arms have the advantage of having more degrees of freedom than a CMM, which has five degrees of freedom. However, none of these algorithms generates a scan path that covers the entire object, taking all the aforementioned constraints into account, while conforming the specified tolerances. Hence, the novelty of the proposed algorithm is the integration of the measurement uncertainty in order to prove conformance to specified tolerances. This integration is made possible by developments of digital twins of optical measurements systems.

## 4. Algorithm

This section describes the working principle of the proposed scan path generation algorithm. The goal of the algorithm is to fully scan an object while minimizing the number of different orientations and enable testing of the conformance of all the specified tolerances. The general working principle of the algorithm is illustrated in Figure 4. The pseudo-Matlab code of the algorithm can be found in the Appendix, Figure A1.

### 4.1. Step 1: Input

The first step in the process is the manual input of all data required beforehand. Three separate files can be defined. Firstly, a Standard Triangle Language (STL) of the object and its position in the CMM space is required. Secondly, all critical features need to be defined. The critical features are the features with a different tolerance for the general geometrical tolerance of the object. Thirdly, a file containing the scan strategies for each individual critical feature needs to be established as an input. Each scan strategy for a critical feature is able to confirm the task-specific tolerances. All these strategies are determined preliminarily by making use of a digital twin [20]. Figure 5 depicts an object where the STL file and the different scan strategies for all critical features are combined. In this example, the object has six features with a tolerance differing for the general tolerance: five cylinder diameters and one distance between two planes. It can be seen in the figure that the diameter of cylinder 1 can be scanned by joint multiple predetermined combinations; for example a scan strategy with following joint combinations: A = 15° and B = −180°, −90°, 0° and 90°. Joint A and B operate as depicted in Figure 2.

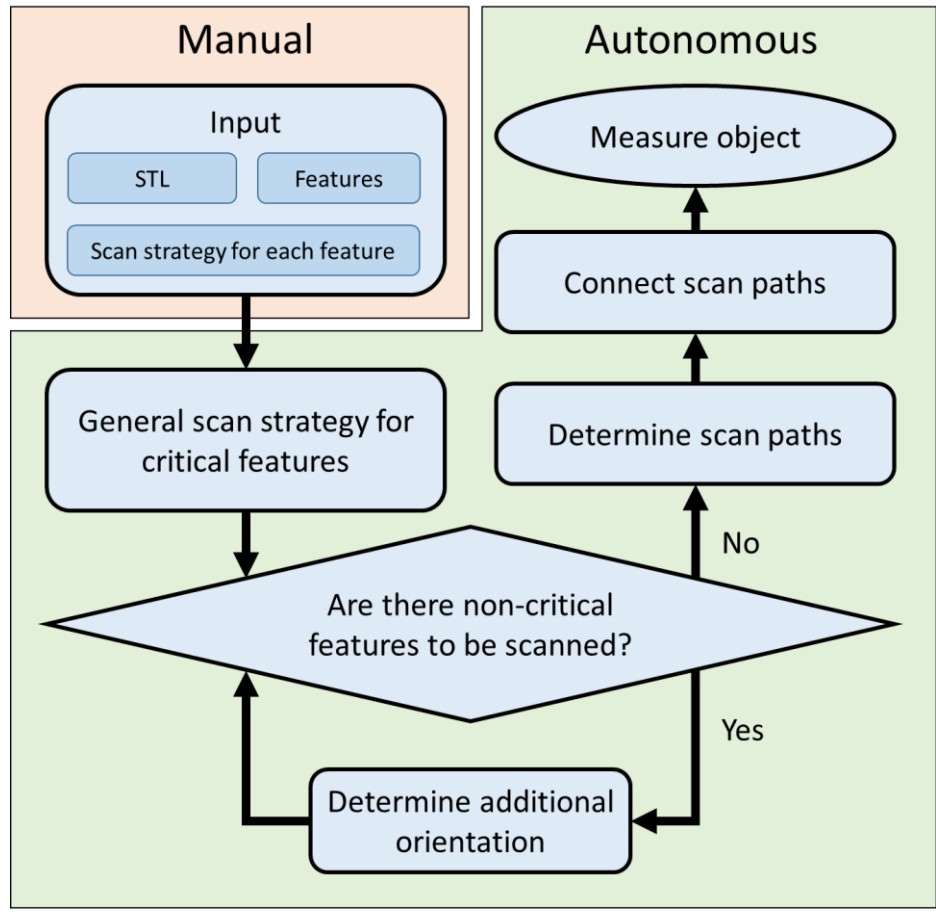

**Figure 4.** Working principle of the path planning algorithm.

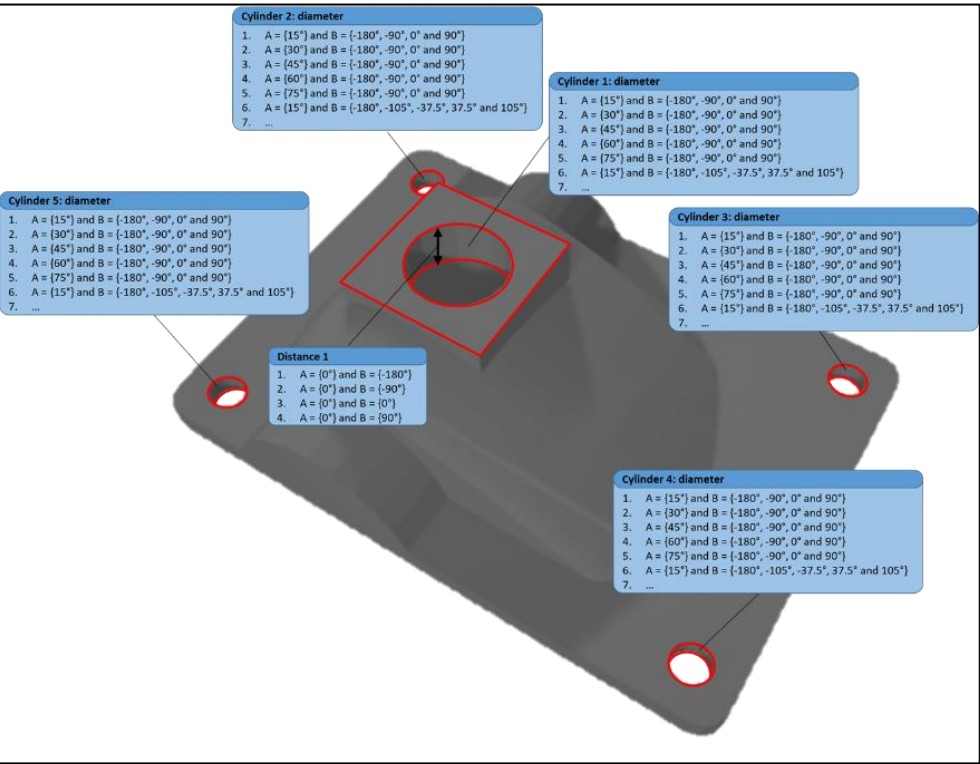

**Figure 5.** An STL file combined with different scan strategies for the critical tolerance features.

*4.2. Step 2: General Scan Strategy for Critical Features*

This step generates the optimal combination of orientations to measure the critical features with a task-specific tolerance, based on the manually determined scan strategies for each critical feature. While conforming the specified tolerances, areas without tolerance are included, if possible, with the orientations determined in this step. A path for the non-critical surface areas, which are not yet scanned, is determined in later steps.

### 4.2.1. Step 2a: Minimum of Different Orientations

Based on the input file with the different scan strategies for each critical feature, the combinations with minimal different orientations are determined. The number of orientations is minimized, since changing of orientation is time-consuming, especially when the LLS needs to move to a safety position to avoid collision, as discussed in Step 5.

All the combinations of the different scan strategies for all critical features are generated. For each combination, the number of unique orientations is determined. If there is only one combination containing the minimum number of orientations, this combination is selected and the algorithm proceeds with Step 3. However, multiple combinations often contain the minimum number of orientations. In that case, the next step is Step 2b, where the total measurable surface for each orientation is calculated and the maximum is selected.

### 4.2.2. Step 2b: Calculate the Coverable Surface Per Orientation

The total coverable area for each orientation part of the different combinations selected in the previous step is calculated. A surface is coverable if it complies with all prerequisites, i.e., maximum angle of reflection, minimal scan distance, position within the CMM space and reachability. The angle between the laser beam and the normal on the surface can be determined based on the STL file.

In order to determine the reachability of a surface, the ech individual triangle of the STL, which complies to the maximum angle constraint, is verified. The working principle to verify the reachability is depicted in Figure 6 and works as follows: (1) the first triangle is selected; in this case triangle a. (2) This triangle forms the base of the view space in the direction of the view direction. (3) Every part of the object within the established view space is determined. These parts are projected according to the view direction on triangle a. In Figure 6(3), the red part is blocked for the LLS by the object and the green part is reachable. (4) This process is repeated for every triangle until the reachability is determined for the specified view direction. The total coverable area for the projected laser light are represented in green. Afterwards, this process is performed to verify the cover limited by the reflected laser light.

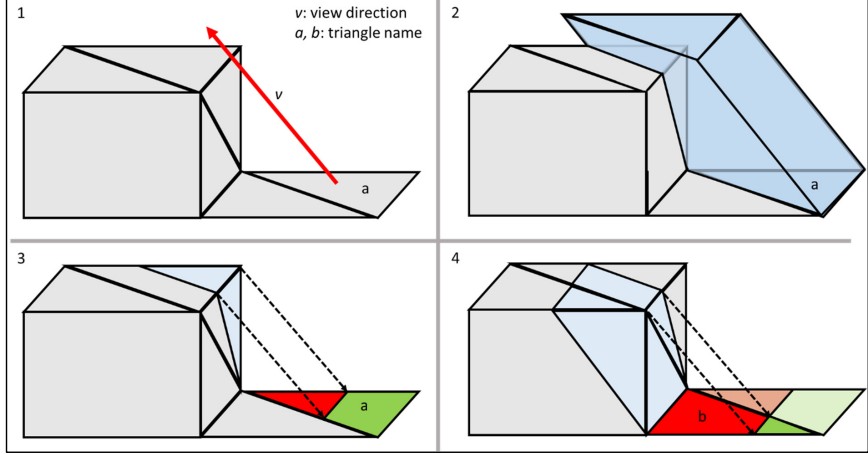

**Figure 6.** Working principle to verify the reachability of a triangle of the STL model. (**1**) The triangle and the corresponding view direction are selected. (**2**) Determination of the view space. (**3**) Determination of reachable surface. (**4**) Repetition of the process for the next triangle.

### 4.2.3. Step 2c: Select Combination with Largest Scanned Surface

For each combination with the minimal orientations, the total coverable area for each orientation is known based on Step 2b. The combination which allows for coverage of the largest surface of the object is selected. If there are still zones of the object which are not covered, the algorithm proceeds with Step 3. Otherwise, Step 3 is skipped by the algorithm.

### 4.3. Step 3: Determine Additional Orientation

This step is similar to Step 2. However, the combinations to prove conformance to the specified tolerances of the critical features are replaced by a list of all the possible orientations of the aph, excluding the orientations selected in Step 2. For each orientation, the total coverable area is determined and the orientation with the maximum coverage is selected. This step is repeated until all surface areas are appointed to an orientation or when the areas are proven to be uncoverable by the LLS.

### 4.4. Step 4: Determine Scan Paths

For each view direction, the corresponding scan paths are determined. This process is illustrated in Figure 7. Firstly, all of the surfaces, which are covered for a specific view direction, are selected. Secondly, these surfaces are divided in depth levels. The LLS has a depth of view and not all areas can be covered with the same scan depth. Therefore, the first depth level starts at the minimum height for the view direction. The second depth level starts one depth of view higher than the first depth level. This is repeated until the maximum height is reached. Thirdly, each depth level is projected along the view direction on a plane. Lastly, the scan paths are established on this projection. The direction of the scan paths corresponds to the direction of the LLS for the specific view direction, i.e., the direction from the sensor of the LLS to the laser source. Each covered area of a scan path has some overlap with the following scan path to avoid under-scanning.

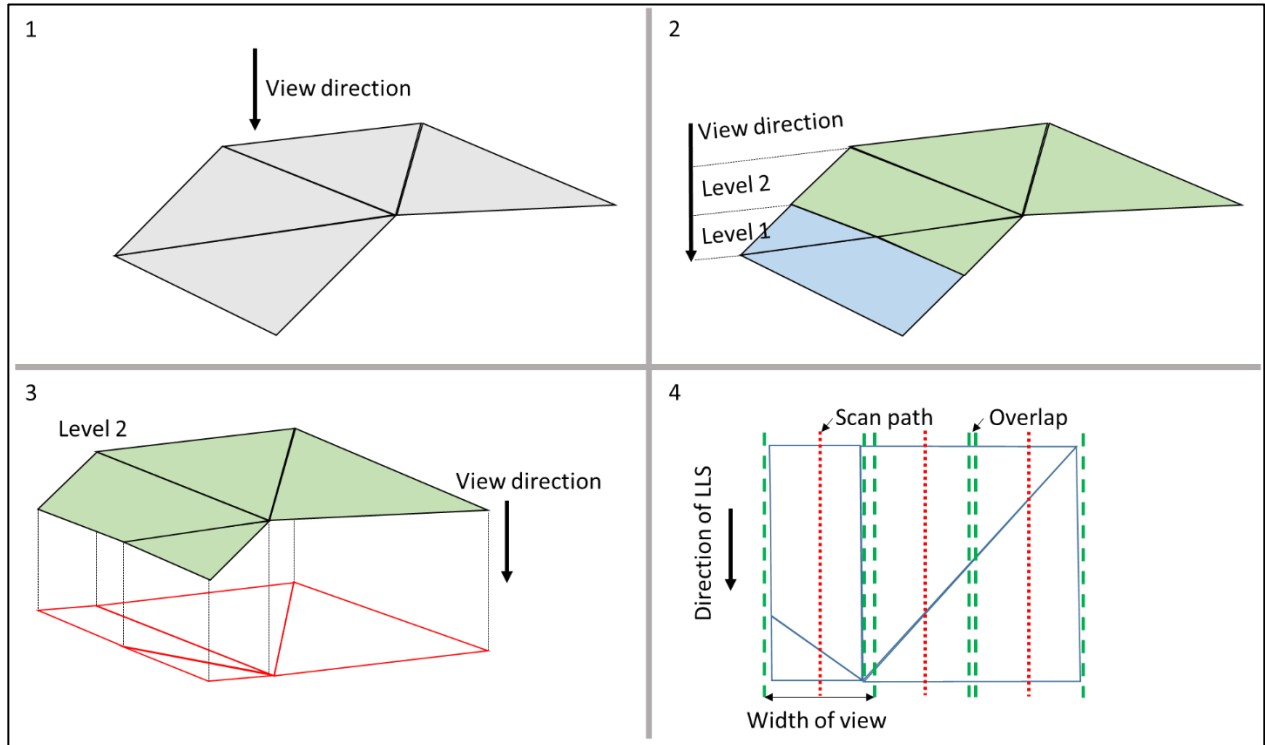

**Figure 7.** Determination of scan paths for a specific view direction. (**1**) Select total surface for a specific view direction. (**2**) Divide surface in different scan depths. (**3**) Projection of each scan depth on to plane. (**4**) Determination of scan paths.

### 4.5. Step 5: Connect Scan Paths

When the scan paths are determined, there must be a collision-free trajectory when moving from the end of one scan path to the start of another scan path. Collisions between the scanner and the object or the CMM need to be avoided. Therefore, the rotation space and movement space are calculated. This is the space that the LLS requires to perform a reorientation or a movement. Figure 8 illustrates both of these spaces. When one of these spaces interferes with the object or the boundaries of the CMM space, a collision is imminent.

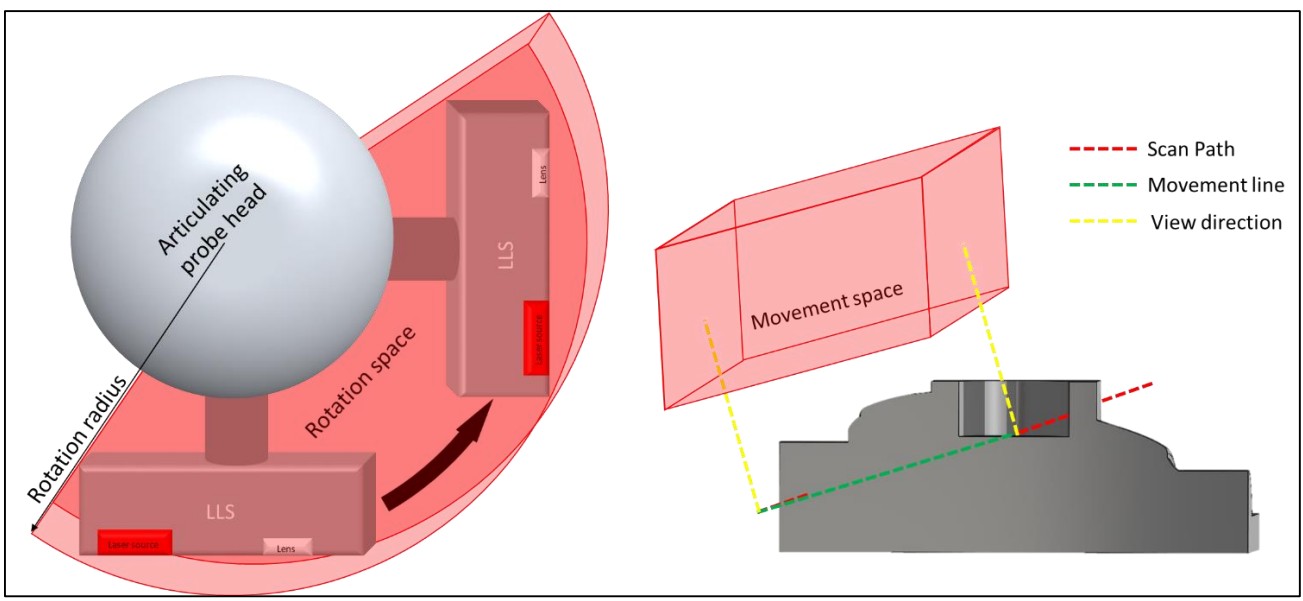

**Figure 8.** The rotation space (**left**) and the movement space (**right**) of the LLS.

In case of an interference of the rotation space, the CMM moves the LLS to a safety position where it can rotate freely. The safety position is in the CMM space above the current position of the LLS with a height of one rotation radius above the object, as illustrated in Figure 8. This movement path towards the safety position is also verified with the collision detection algorithm.

In case of an interference of the movement space and the object, the movement line is adapted. Since the CMM repositions the LLS in a straight line, the movement line is also a straight line of the combinations of multiple straight lines. When a collision is detected, the movement line is shifted contrary to the view direction. The movement line translates so the movement space surpasses the maximum of the object in perspective of the view direction. This situation is illustrated in Figure 9.

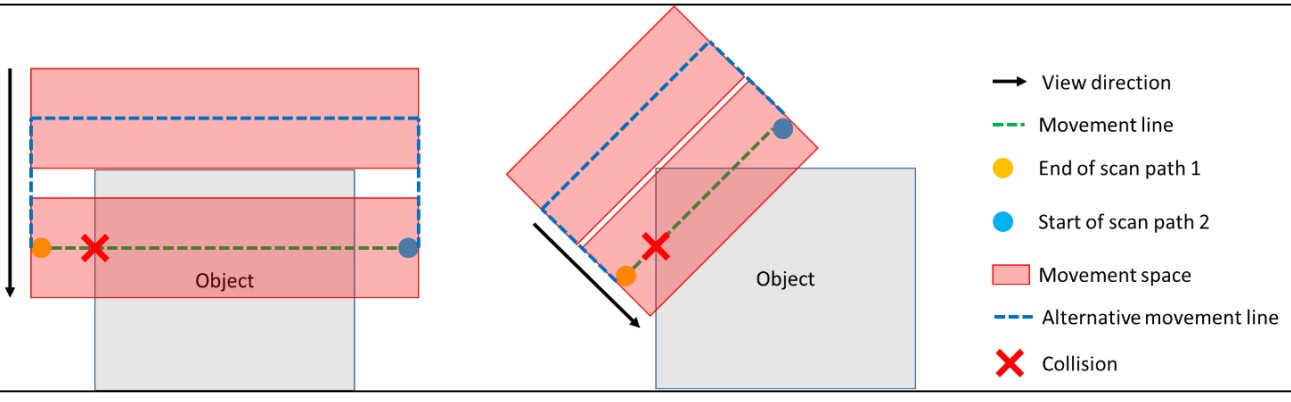

**Figure 9.** Adaption of the movement line in case of a collision.

*4.6. Step 6: Measure Object*

When the scan paths and movement lines are determined, the PC-DMIS code is created based on the generated information and the object is measured. Dimensional Measurement Interface Specification (DMIS) is the code that most CMMs run on.

## 5. Validation

For the validation of the path planning algorithm, the scan path for two objects is determined. The two objects have different surface characteristics, depicted in Figure 10. As a consequence of the different surfaces, the maximum angle of reflectance differs, thus the generated scan path will be different as well. One surface is matte to avoid reflective glare and has a maximum angle of reflectance of 75°. The other surface is milled aluminum, which strongly reflects the laser light, limiting the maximum angle of reflectance to 30°. The aluminum part has no small drilled holes. Therefore, the uncertainty analysis is validated on the matte object. However, the surface of the aluminum part still allows the validation of complete coverage by the LLS.

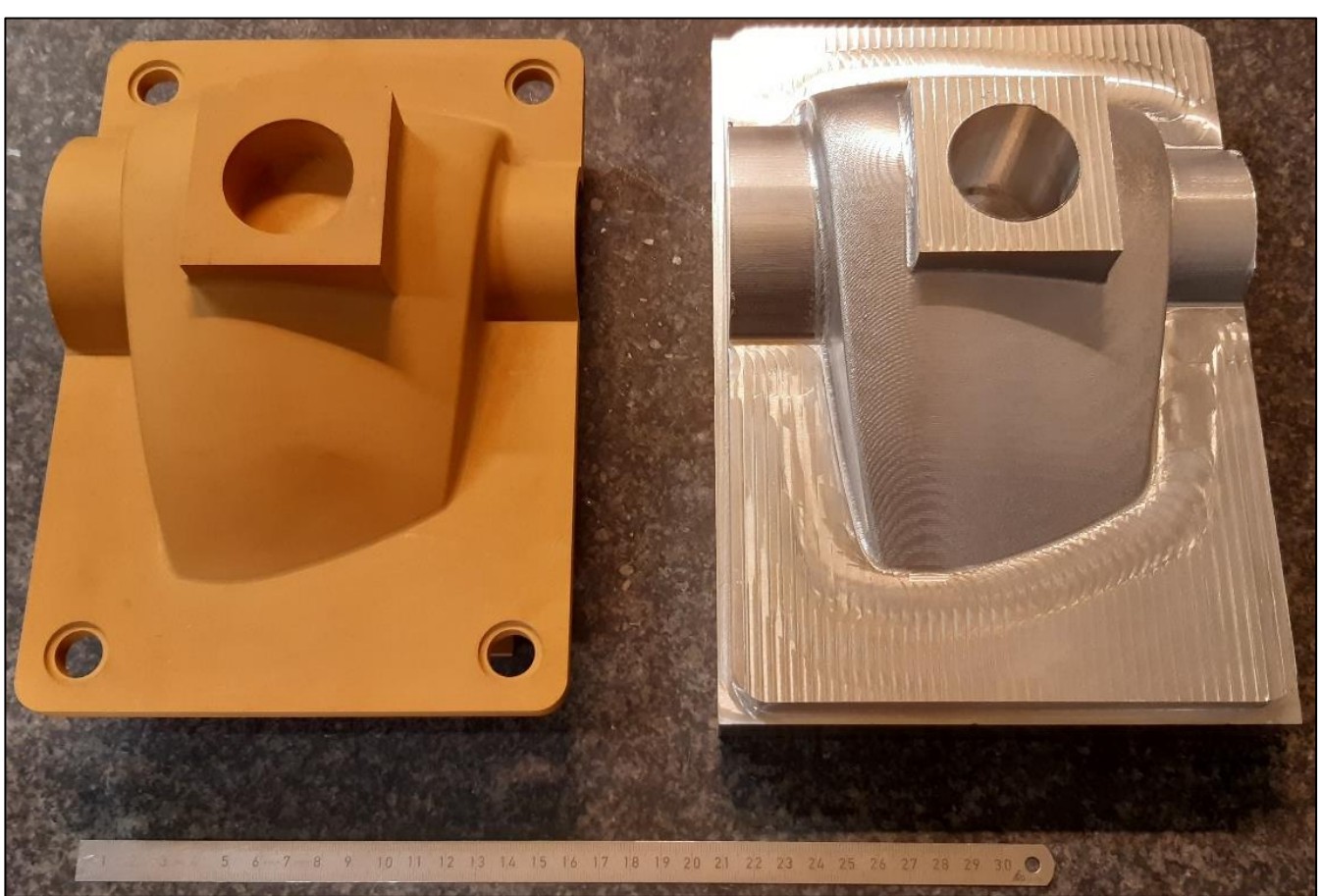

**Figure 10.** The validation objects with different surface characteristics, with the matte object on the left and the reflective object on the right. The measurement volume is about 250, 180 and 95 mm in the x-, y- and z-direction, respectively.

The objects have the specified tolerances as discussed in Step 1 and illustrated in Figure 5. The specific tolerances for each feature are listed in Table 1. The diameter of five inner cylinders and the distance between two planes need to be determined. Different scan strategies are evaluated with the digital twin. The correctness of the uncertainty determination is evaluated by comparing the estimated uncertainty with the measured repeatability of the cylindrical critical features. The measured repeatability is an estimate of the measurement uncertainty. However, the measurements are performed on one specific

measurement device, the MC16 Coord3 16.10.8 and the Nikon Metrology N.V LC60Dx LLS. As a consequence, the uncertainty due to the geometric kinematical errors of the CMM, assumed by the digital twin, are not present in the measured repeatability. Therefore, two types of uncertainty are calculated: the minimal uncertainty, i.e., the uncertainty of the aph and the LLS, and the maximal uncertainty, thus including the kinematical errors of the CMM. The measured repeatability should be between those two values for validation. The repeatability is determined for the diameter of five inner cylinders with 10 measurements for each hole. Hole 1 is the large inner cylinder with a reference diameter of 40.043 ± 0.001 mm, as determined on the CMM using a tactile probe. The diameters of the small holes 2–5 have a reference nominal diameter of 15 mm; the reference diameters are listed in Table 1. Each hole is measured on four sides with an angle of 45°. Figure 11 depicts the comparison of the five holes. For all holes, the measured repeatability is within the values of the minimum and maximum virtually estimated uncertainty. The measured repeatability is larger than the minimum uncertainty due to the fact that some error contributors are not included in the digital twin. For example, the hysteresis of the CMM is present in the repeatability tests, but is not included in the digital twin. Furthermore, the limited number of experiments have an influence on the determined repeatability, i.e., the variation of the repeatability of holes 2–5.

**Table 1.** The specifications, measurements and simulations for all critical features.

| Feature | Diameter Hole 1 | Diameter Hole 2 | Diameter Hole 3 | Diameter Hole 4 | Diameter Hole 5 | Distance between Planes |
|---|---|---|---|---|---|---|
| Nominal value/mm | 40 | 15 | 15 | 15 | 15 | 20 |
| Tolerance | G8 | H5 | H5 | H5 | H5 | m |
| Reference value/mm | 40.011 | 15.004 | 14.999 | 14.991 | 15.001 | 20.160 |
| 95% confidence interval/mm (reference) | [40.009; 40.013] | [15.032; 15.037] | [14.977; 14.982] | [14.949; 14.954] | [14.999; 15.004] | [20.158; 20.161] |
| Measured value/mm | 40.047 | 15.007 | 15.006 | 15.006 | 15.007 | 20.162 |
| 95% confidence interval/mm (experimental) | [40.045; 40.048] | [15.006; 15.008] | [15.005; 15.007] | [15.005; 15.007] | [15.006; 15.008] | [20.161; 20.164] |
| 95% minimal confidence interval/mm (virtual) | [40.046; 40.048] | [15.006; 15.008] | [15.005; 15.007] | [15.005; 15.007] | [15.006; 15.008] | [20.162; 20.163] |
| 95% maximal confidence interval/mm (virtual) | [40.045; 40.048] | [15.006; 15.009] | [15.005; 15.008] | [15.005; 15.008] | [15.006; 15.008] | [20.161; 20.164] |
| Conformance probability | 95% | 89% | 99% | 99% | 93% | 100% |

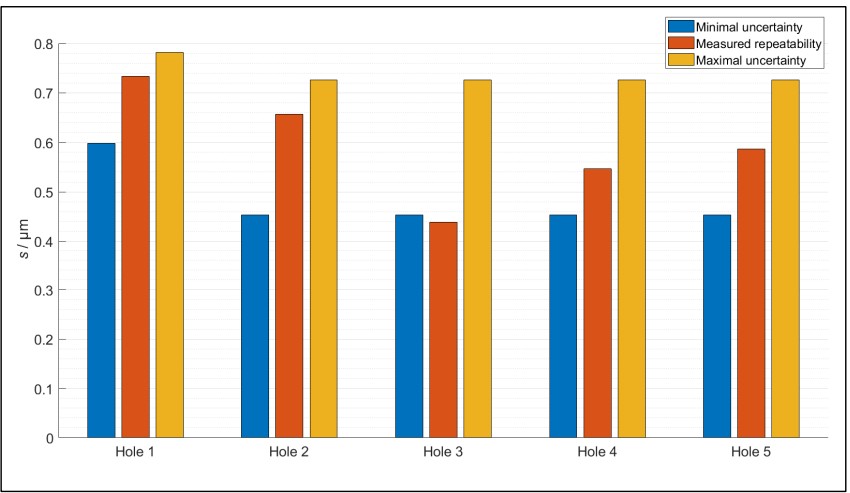

**Figure 11.** The measured error on the diameter and the virtually determined uncertainty intervals.

Figure 12 illustrates the reference measurements, specification zone and virtual measurements on the average measured error by the LLS of the diameter of hole 1. Figure 13 depicts a section view of Figure 12 for clarity. Due to the systematic error, the probabilities for conformance to the specified tolerance are 98% for the minimal uncertainty, and 95% for

the maximal uncertainty. In order to improve this number, a different scan strategy with a lower uncertainty interval has to be applied. It must be noted that it may well happen that the uncertainty interval overlaps with the tolerance zone, and ISO 14253-1 gives rules and procedures for this situation. A general a rule-of-thumb is that a measurement method should have an uncertainty that is smaller than 20% of the tolerance interval, which appears to be the case here. Similarly, as depicted in Figure 12, the other features are validated. The measured values, i.e., tactile and optical, are listed in Table 1, along with the experimentally obtained 95% confidence interval as the simulated confidence interval.

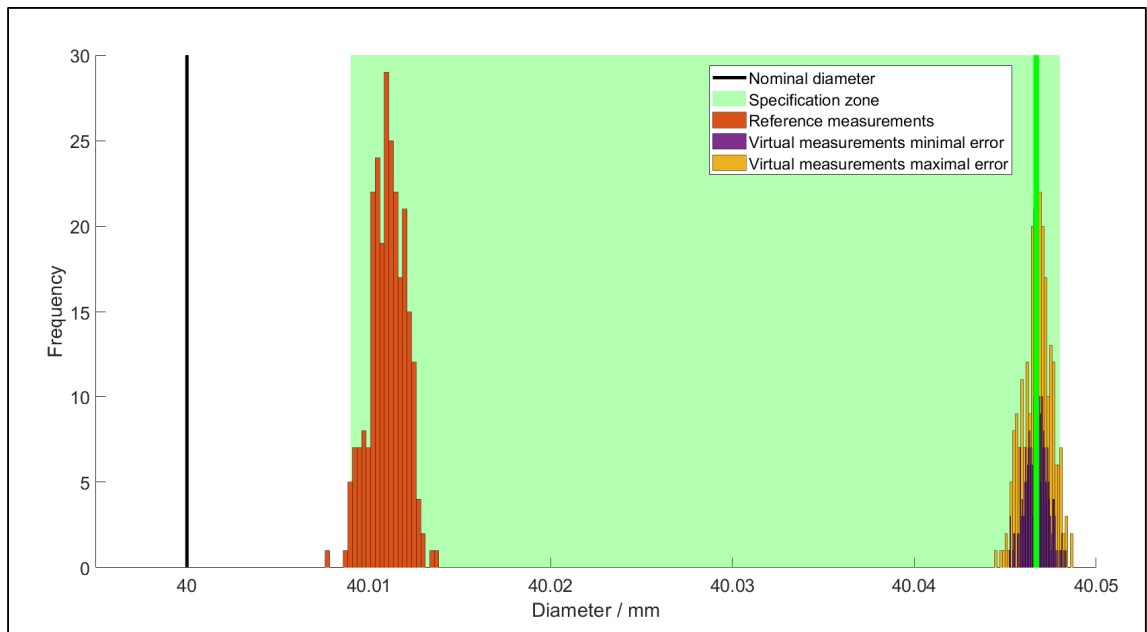

**Figure 12.** The measurement uncertainty on the measured diameter and the influence on the conformance according to a specified tolerance.

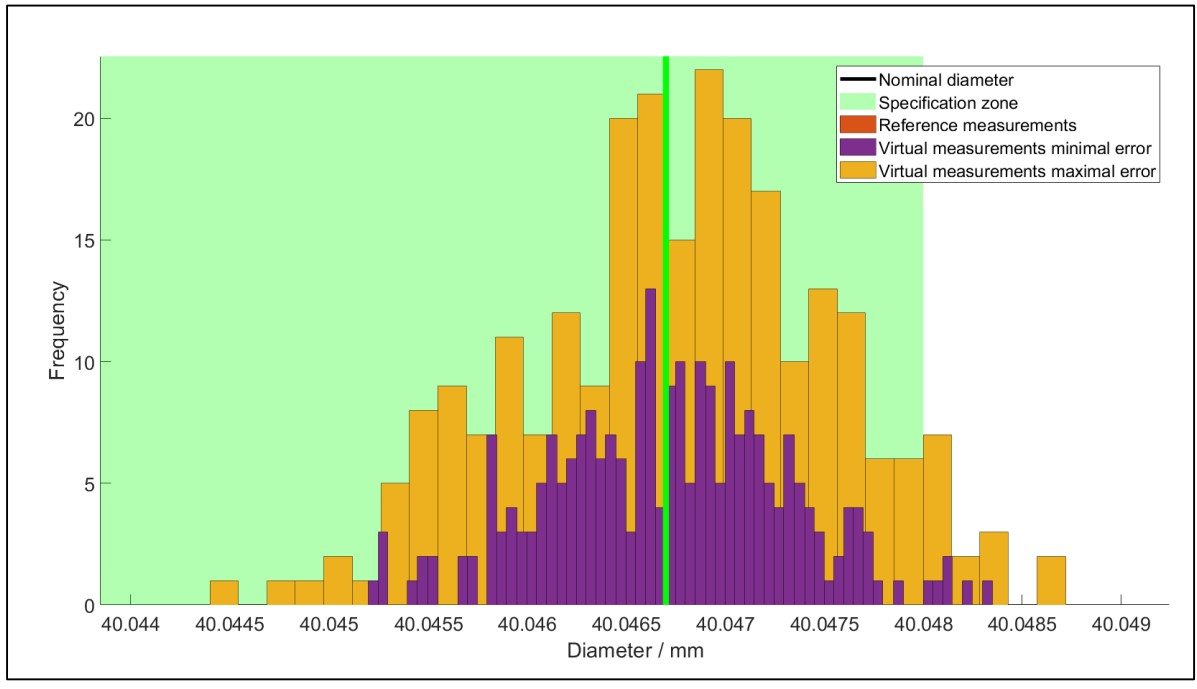

**Figure 13.** Section view of Figure 12.

Figure 14 depicts an STL (for visibility) of the point cloud measured by the generated scan path for each orientation of the matte object. The measurements taken in these four orientations merged together are able to cover the entire surface. Figure 15 shows the generated point cloud of the matte surface on the left. Figure 15 also shows the experimentally generated data of the reflective object on the right. For this object, the orientations for the evaluation of the features were not sufficient to cover the entire surface and additional orientations are required. It can be noted from Figure 15 that the density of the point cloud is lower due the reflective character of the object. From Figure 15, it can be concluded that both objects are maximally scanned, thus the algorithm generates a correct scan path.

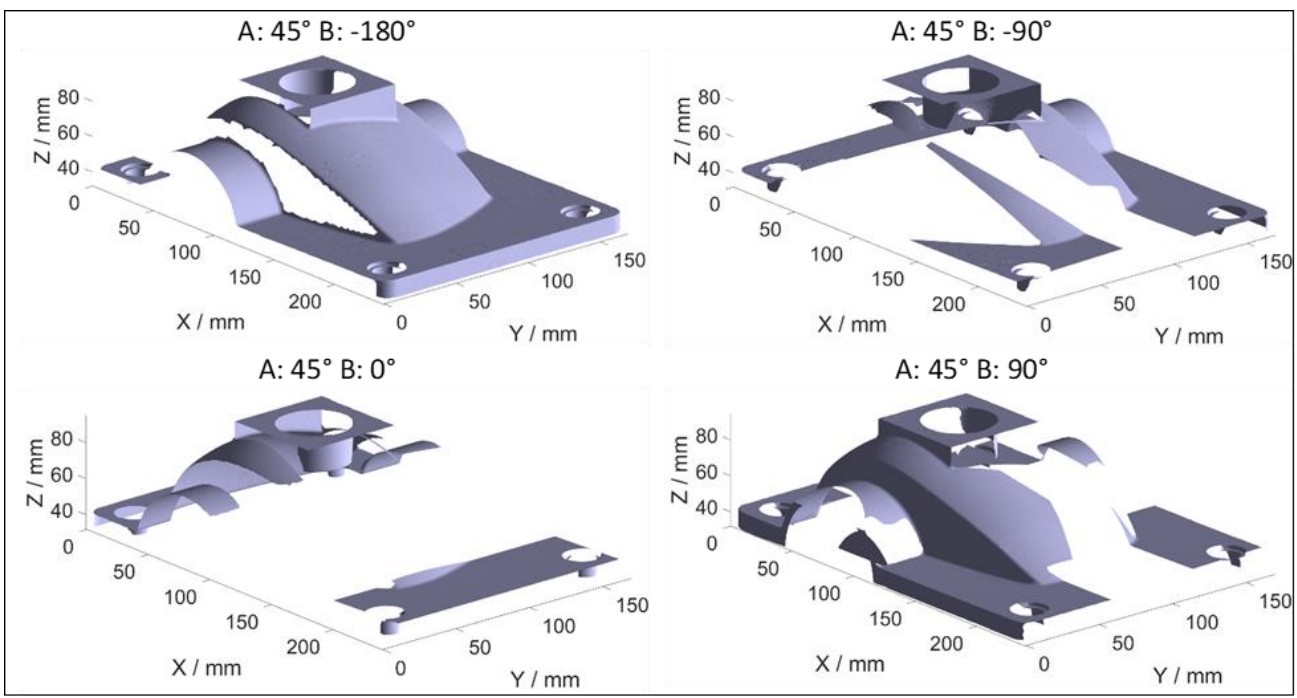

**Figure 14.** The generated point clouds for the evaluation of the features of the matte object with four different orientations.

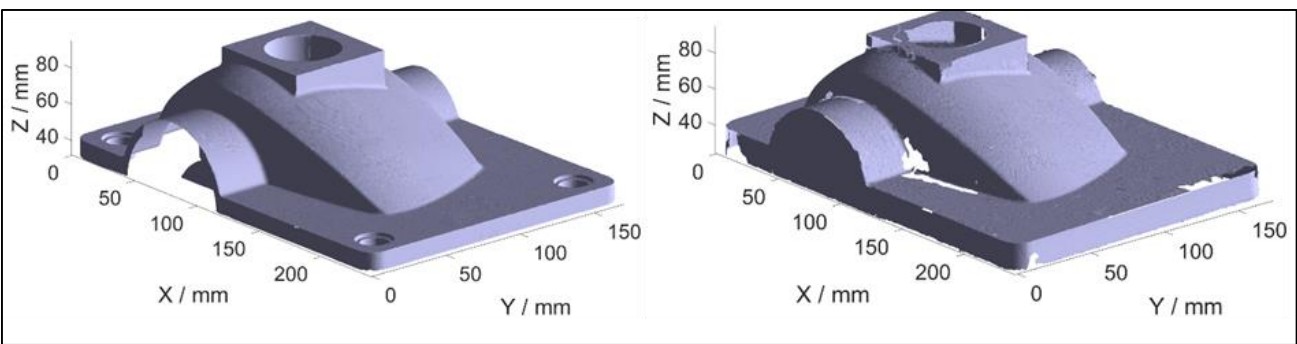

**Figure 15.** The experimental data of the matte object (**left**) and the experimental data of the reflective object (**right**).

The objects depicted in Figure 10 have low complexity in terms of reachability of the surface. In order to prove the algorithm's capacity to deal with more complex geometries, an Archimedean screw is virtually measured. The screw is depicted in Figure 16. The maximum angle of reflectance is supposed to be 40°, making it difficult to reach the inner parts of the screw. Figure 16 depicts the measured point cloud obtained through the algorithm. As can be seen in the picture, the inner screw cannot be scanned due the

blockage by the upper parts in combination with a small maximal angle of reflectance. Furthermore, the algorithm was able to predict these unmeasurable areas and did not create additional scan paths to measure these areas.

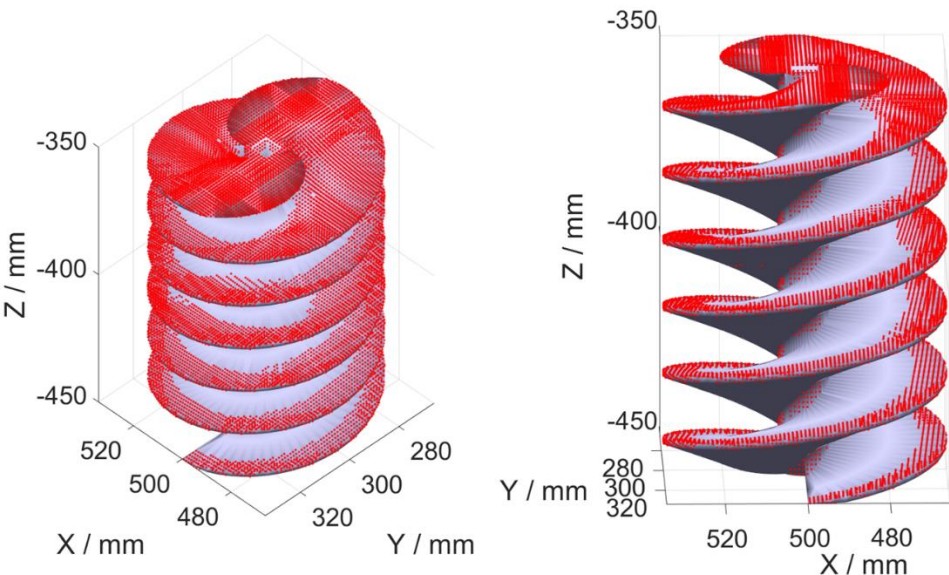

**Figure 16.** A virtual measurement of an Archimedean screw with the measured points in red.

## 6. Conclusions

This article gives an overview of the constraints of the optical measurement system and discusses the shortcomings of the state-of-the-art algorithms. Additionally, an algorithm is proposed to offer a solution to the short-comings of the existing algorithms. The proposed algorithm takes into account the constraints and acknowledges the importance of the uncertainty budget when performing QAs. This novel approach in path planning provides the ability to prove conformance to specified tolerances after the measurement. The task-specific measurement uncertainty determination is enabled due to recent developments for digital twins for laser line scanners. The algorithm is validated by scanning the object with different surface characteristics experimentally and virtually with an in-house developed digital twin. Potential applications for the algorithm can be found in the manufacturing industry, which manufacture unique or small product batches, such as the additive manufacturing industry, or the aerospace industry, where 100% quality assurance is required.

Future work could consist of creating a manner to integrate the possible scan strategies of specified features in the STL model, similar to how a PMI-file integrates non-geometric attributes of an object with a three-dimensional (3D) model. This will make the system in accordance with the principles of Industry 4.0. Furthermore, the algorithm could be improved by integrating better path finding algorithms to avoid a collision and find a shorter path between different scan tracks. The proposed algorithm does not yet generate the absolute optimal scan paths in terms of scan time minimization. Scan path adaptions can be integrated on the fly in order to lower the uncertainty interval when 100% conformance cannot be given; this situation is illustrated Figure 12.

This algorithm was programmed in MatLab as a proof of concept. In order to optimize the algorithm's calculation time, it is recommended to program in another computer language. The authors assume that the proposed algorithm in the current state is outperformed by the state-of-the-art in terms of calculation time. Once the algorithm is translated to another computer language, a comparison with the state-of-the-art can be made.

**Author Contributions:** Writing—original draft preparation, M.V.; writing—review and editing, H.H. aitjema and W.D. All authors have read and agreed to the published version of the manuscript.

**Funding:** This research received no external funding.

**Institutional Review Board Statement:** Not applicable.

**Informed Consent Statement:** Not applicable.

**Data Availability Statement:** The data that support the findings of this study are available upon reasonable request from the authors.

**Conflicts of Interest:** The authors declare no conflict of interest.

## Appendix A

**Figure A1.** The pseudo-MatLab code for the autonomous scan path generation.

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
