# Peer review of "Uncertainty-Based Autonomous Path Planning for Laser Line Scanners"

_2673-8244, doi:10.3390/metrology2040028_

Round 1

Reviewer 1 Report

This paper presents an Autonomous path planning for laser line scanners. The topic is interesting, and the application of the proposed algorithms can be very useful in the industry.

Some details need to be attended to enhance the paper presentation.

General comments:

1.       Some references are old and should be updated for more recent work.

2.       This document can be improved by adding a comparison of the present work against similar algorithms. A comparison of memory, execution time, and complexity is recommended.

3.       It is recommended to add more case studies with more complex pieces and different types of lighting to see their effect on the performance of the methodology.

4.       The description of the algorithm employing pseudocode would be very useful for the reader.

Reviewer 2 Report

Dear authors,
In the beginning, I would like to congratulate you on the interesting idea for a paper. However, the aim of a scientific paper is to present some new ideas. In this case, it is asked to present stronger the novelty of the research.
A)    General remarks
1    Article is clearly written and easy to follow. The authors give relevant references which are linked to their study.
2     The abstract is well written introducing the basic overview of the paper. It is also written in a way that even a person not familiar with the topic can understand what the authors are proposing in their research. However, the novelty component of the paper is not presented in the abstract. This must be improved.
3.    The introduction provides the basic background of the paper and an overview of the methods used by the authors.
It is suggested to introduce some abbreviations which are commonly used e.g. Cyber-Physical Systems (CPS). Also, not only in the introduction but in the whole article the authors are using abbreviations which were not introduced (CAD, CNC…). Those names are commonly used but it would be more professional to give the full name when introduced for the first time. Please check the whole article carefully to address both matters.
Line 28 authors are using Quality inspection which is too general. Please be specific as this is for quality assurance (QA) or quality control (QC) or both.  Give some examples of contact and none- contact systems like visual (high-speed cameras e.g doi: 10.3390/s21196643), optical (3D laser vibrometry e.g DOI: 10.1109/IDAACS53288.2021.9661060, Digital Image correlation e.g https://doi.org/10.1016/j.rineng.2020.100109) and others.  
4. The methods are clearly presented.
5. The results are clearly presented.
6. Concluions are sufficient. However, again the novelty component is not presented as well as a better explanation of possible industrial applications.
B)    Item remarks
The figures are correct and are of good quality. No improvements are necessary.

C)    Conclusions
In the current form, the reviewer is asking for minor revisions and especially asks to improve the statement of novelty.

Round 2

Reviewer 1 Report

The work presented is interesting. The comments suggested to the authors have been appropriately answered. I recommend this document for publication.